# Advances in Brain Stimulation, Nanomedicine and the Use of Magnetoelectric Nanoparticles: Dopaminergic Alterations and Their Role in Neurodegeneration and Drug Addiction

**DOI:** 10.3390/molecules29153580

**Published:** 2024-07-29

**Authors:** Silvia Giménez, Alexandra Millan, Alba Mora-Morell, Noa Ayuso, Isis Gastaldo-Jordán, Marta Pardo

**Affiliations:** 1Department of Psychobiology, Universidad de Valencia, 46010 Valencia, Spain; silviauv00@gmail.com (S.G.); noahi@alumni.uv.es (N.A.); 2Department of Neurobiology and Neurophysiology, Universidad Católica de Valencia San Vicente Mártir, 46001 Valencia, Spain; alexamillan31191@gmail.com; 3Faculty of Biological Sciences, Universidad de Valencia, 46100 Valencia, Spain; almomo2@alumni.uv.es; 4Psychiatry Service, Doctor Peset University Hospital, FISABIO, 46017 Valencia, Spain; isis.gastaldo@fisabio.es; 5Interuniversity Research Institute for Molecular Recognition and Technological Development (IDM), 46022 Valencia, Spain

**Keywords:** brain stimulation, nanomedicine, dopamine, neurodegeneration, addiction, magnetoelectric nanoparticles

## Abstract

Recent advancements in brain stimulation and nanomedicine have ushered in a new era of therapeutic interventions for psychiatric and neurodegenerative disorders. This review explores the cutting-edge innovations in brain stimulation techniques, including their applications in alleviating symptoms of main neurodegenerative disorders and addiction. Deep Brain Stimulation (DBS) is an FDA-approved treatment for specific neurodegenerative disorders, including Parkinson’s Disease (PD), and is currently under evaluation for other conditions, such as Alzheimer’s Disease. This technique has facilitated significant advancements in understanding brain electrical circuitry by enabling targeted brain stimulation and providing insights into neural network function and dysfunction. In reviewing DBS studies, this review places particular emphasis on the underlying main neurotransmitter modifications and their specific brain area location, particularly focusing on the dopaminergic system, which plays a critical role in these conditions. Furthermore, this review delves into the groundbreaking developments in nanomedicine, highlighting how nanotechnology can be utilized to target aberrant signaling in neurodegenerative diseases, with a specific focus on the dopaminergic system. The discussion extends to emerging technologies such as magnetoelectric nanoparticles (MENPs), which represent a novel intersection between nanoformulation and brain stimulation approaches. These innovative technologies offer promising avenues for enhancing the precision and effectiveness of treatments by enabling the non-invasive, targeted delivery of therapeutic agents as well as on-site, on-demand stimulation. By integrating insights from recent research and technological advances, this review aims to provide a comprehensive understanding of how brain stimulation and nanomedicine can be synergistically applied to address complex neuropsychiatric and neurodegenerative disorders, paving the way for future therapeutic strategies.

## 1. Innovations in Brain Stimulation and Their Applications in Psychiatry and Neurodegeneration

Neurodegenerative and psychiatric disorders represent a significant portion of the causes of mortality worldwide, and most importantly, include a detrimental impact on patients’ quality of life. Additionally, dealing with these disorders involves billions of dollars across the world in every country’s healthcare system. The heterogeneity of the brain and, therefore, the complexity of psychiatric and neurological pathologies, in addition to the limited efficacy of the available treatments and the big percentage of clinical trial failure, pushes for the need of and investment in innovative resources.

Currently, we have over 60 million patients worldwide facing neurodegenerative disorders, including Alzheimer’s Disease (AD), Parkinson’s Disease (PD) and Amyotrophic Lateral Sclerosis (ALS). The rising prevalence and incidence of neurodegenerative disorders highlight the critical necessity for intensified research, improved diagnostic methodologies and the development of more efficacious therapeutic interventions. Presently, most available treatments are palliative, primarily aimed at symptom management rather than offering a cure for these diseases. Therefore, there is still a need for additional research focused on a better understanding of the alterations mediating these devastating disorders.

We are currently in an evolving era where non-pharmacological approaches for the treatment of neurological conditions are on the rise, either in combination with well-established drug treatments or on their own [1,2].

As far as innovative treatments, electric stimulation has a long history of medical applications. Deep brain stimulation (DBS) is a well-established non-pharmacological treatment approved for its use in a variety of conditions, such as depression, compulsive disorders and PD [3,4]. It involves the use of electrodes that are placed near deep structures of the brain and connected to a wire and pulse generator, which will fire the electrodes when instructed by a computer [5]. Additionally, repetitive transcranial magnetic stimulation (rTMS), approved for unipolar depression and obsessive–compulsive disorder (OCD) [6], is capable of inducing small electric currents in a contained area via the application of a rapidly changing magnetic field to the superficial layers of the cerebral cortex. Less invasive neuromodulation techniques are being used, including transcranial direct current stimulation (tDCS), whose way of working is by direct weak electric currents through scalp electrodes to induce changes in cortical excitability near the anode or the cathode, respectively [7,8]. In parallel, ultrasound approaches have been ‘rediscovered’ as an advanced engineering technique, which has unprecedented accuracy for reaching small areas of the brain [9]. Additionally, ultrasound is not sensitive to changes in conductivity, which makes this approach unique from other methods that require significant changes in normal conductance [10]. Currently, Transcranial Pulse Stimulation (TPS), an ultrasound-based technique that uses ultrashort pressure pulses (3 μs), has been approved for AD treatment [10].

Going even further, brain–computer interfaces (BCIs) have provided great potential in the treatment of a wide array of neurological disorders [11,12,13] via new output pathways [14]. Since 1973, progress has slowly been made to bring the technology to a point where it can potentially be used in a clinical environment [15]. Indeed, there are several BCI technologies that, as of 2021, are being tested in clinical trials, targeting different neurological conditions [16].

As of today, perhaps the most well-known BCI technology is Neuralink, Elon Musk’s foray into medical sciences. This technology is composed of ultra-fine polymer probes or “threads”, a neurosurgical robot and customized high-density electronics. The threads record neuronal activity and send it to the custom chip, which decodes the electrical signal from the brain and transmits it to a digital device. Testing of this technology in Long–Evans rats allowed for the recording of widespread neuronal activity [17]. Testing for the Neuralink implant has also been conducted in primates, and even in the first human. However, there is not much information readily available on the results of either trial. Neuralink reported that the monkeys were able to play Pong without the use of physical controllers, although the Physicians Committee for Responsible Medicine (PCRM) raised an issue on the treatment and death of the primates [18,19].

As for the first human patient, Noland Arbaugh, he was able to utilize Neuralink’s implant to control digital devices. Recently though, it seems as if most of the implant’s threads have become unresponsive, limiting his control of external devices [20]. Even so, the FDA has green-lit the implant for clinical trials, and the company is currently recruiting participants for their N1 implant and R1 robot. If successful, it would allow tetraplegia patients to control digital devices through their implants [21].

However, according to Zhang & Dai (2024) [22], the day after Tesla CEO Elon Musk announced his successful implantation, China’s Tsinghua University announced that they had successfully rehabilitated the first BCI patient with their wireless device, meeting the highest safety standards. Interesting discussions are taking place in regard to the invasiveness of current BCIs with promising outcomes. Additional BCI implants continue to gain attention, such as Synchron with their new endovascular electrode, the Stentrode, which is currently being used in the clinic and enables neuroprosthetics, neuromodulation and neurodiagnostics [23,24].

How far have we arrived in regard to the use of those innovative approaches in neurodegeneration treatment? Neurofeedback (neurorehabilitation using BCIs) provides real-time information of neural activity that is made available to the patient, therefore, allowing for learning how to control neural activity and, consequently, target symptomatology. Limited and promising data demonstrated that BCIs improved independence and autonomy in ALS patients, despite the progressive decline of their neuromuscular functions [13,25]. Several studies tested BCIs in PD, where BCIs showed improved locomotor ability and alleviated some additional symptoms [11,12]. As recently reviewed [26], a BCI requires additional studies before we can implement it in clinical setup. Neuralink also proposes to target neurodegenerative diseases, but lots of questions are still raised in regard to the safety and efficacy of the device due to the requirement of neurosurgical robots to implant the device’s magnitude of electrodes [27].

Although promising, most of the innovative approaches mentioned above still have limited data supporting their use for neurodegenerative disorders. Even though TMS has been around for decades, it is still only approved for psychiatric conditions such as depression [28] and OCD [29]. DBS remains the “gold-standard” technique when considering stimulation approaches for brain targeting.

The development of more sophisticated and precise techniques, such as neural interfaces, has opened new frontiers in neurology and neuroscience, providing opportunities to understand the brain and develop therapeutic devices to restore or replace lost functions. A potentially more significant societal contribution lies in elucidating the benefits conferred by these technologies to patients, as well as the underlying mechanisms responsible for these effects. For that purpose, it is mandatory to better understand the electrical characteristics of the brain, which includes identifying the specific brain regions and neurotransmitter pathways involved, which respond to both chemical and electrical inputs. Targeting those neurotransmitters is instrumental in achieving such remarkable therapeutic outcomes.

The expanded sections of this review integrate contemporary insights essential for the prudent application of brain stimulation in complex contexts like neurodegenerative disorders. Our understanding of how electric and magnetic stimulation elicits responses in the brain, along with the biochemical pathways involved in neurodegeneration, remains actively developed. Moreover, identifying the specific brain regions implicated in these processes holds significant importance. Notably, given dopamine’s (DA) pivotal role across various neurodegenerative conditions, particular emphasis is placed on elucidating alterations within the dopaminergic system.

## 2. Brain Stimulation Therapeutic Intervention in Neurodegenerative and Psychiatric Conditions

Due to the complexity of the brain, we continue to uncover underlying mechanisms that mediate specific symptoms of these disorders. The improved characterization of specific targets and their roles within distinct brain regions in pathological conditions could enhance the development and efficacy of innovative treatment approaches.

Several brain stimulation devices are approved for therapeutic applications in neurodegenerative conditions. DBS devices, including those from Medtronic (Dublin, Ireland), Abbott (formerly St. Jude Medical) (Chicago, IL, USA) and Boston Scientific (Boston, MA, USA), are used to treat PD, essential tremor and dystonia. Vagus Nerve Stimulation (VNS) systems, such as the VNS Therapy System by LivaNova, are approved for epilepsy, which can sometimes be associated with neurodegenerative disorders. While TMS devices like the NeuroStar TMS Therapy system and Brainsway Deep TMS system are approved for depression, obsessive–compulsive disorder (OCD) and migraines, they are being explored for potential applications in neurodegenerative diseases like AD and PD. Responsive Neurostimulation (RNS) systems such as the NeuroPace RNS System are approved for epilepsy, with ongoing research into their efficacy for other neurodegenerative conditions. These devices undergo stringent regulatory scrutiny by bodies such as the U.S. FDA to ensure their safety and effectiveness in managing neurodegenerative disorders. Therefore, most of the data to be collected and discussed in the following sections pertain to experimental conditions. Limited clinical data, where available, are also presented for comprehensive analysis.

### 2.1. Therapeutic Targets: Dopaminergic Alterations

The DArgic system, through various circuitries, has been strongly associated with neurodegenerative and psychiatric disorders. DA is a key neurotransmitter involved in numerous processes, including motor function, cognition and reward. The DArgic system has been proposed for decades as a target for neuromodulation to control symptoms in patients with various pathologies. In the following section, we will describe DArgic implication in the most relevant neurodegenerative disorders as well as in drug addiction. In addition, Section 2.2 will focus on the brain areas that are being studied to impact the DArgic system in such disorders.

#### 2.1.1. Amyotrophic Lateral Sclerosis (ALS) Symptomatology and Underlying Mechanisms: Hyperexcitability and Dopaminergic Implication

ALS is a neurodegenerative disorder characterized by progressive motor symptoms and cognitive behavioral changes via its alterations on superior and inferior motor neurons (MNs). The inclusion of pathological TDP-43 aggregations (hallmark of the disease) spread initially from MNs in the motor cortex, spinal cord and brainstem motor nuclei to other brain regions, such as neocortical areas, the cerebellum and striatum [30]. With no cure or even a fully effective treatment available, current stimulation devices have helped its pathophysiological understanding and the development of new diagnostic approaches [31].

Current treatments for ALS are based on pharmacological approaches targeting glutamate excitotoxicity (FDA-approved riluzole), oxidative stress (for example, edaravone), inflammation or therapies trying to clear the TDP-43 formation [32,33]. Due to its complexity, stimulation approaches have also made their way to clinical trials [31], highlighting which brain area they focus on: the motor cortex. TMS offers a promising approach for the treatment of ALS, with the potential to alleviate symptoms and improve the patients’ quality of life [34]. However, some research suggests that DBS may have a potential therapeutic effect in ALS by targeting specific brain regions associated with symptom management or by modulating neuroplasticity and neural networks.

DA and glutamate are two of the main mediators of the characteristic cortical hyperexcitability, an early feature of ALS patients. It affects mostly the corticomotor neurons, the key factor of this devastating neurodegenerative disorder [35,36]. DArgic imbalance in the nigrostriatal and mesolimbic pathways is crucial in ALS pathology [36,37], contributing to motor and non-motor symptoms. Reduced DA levels were found using PET in ALS patients [36] and preclinical animal models [38]. Additionally, decreased D2 receptors in striatal areas were reported in ALS patients [39]. Recently, D2 receptors have been confirmed as modulators of ALS motor neuron excitability in clinical trials [40,41]. Notably, in the ALS mouse model, 50% of the DArgic neurons in the ventral tegmental area (VTA) were lost, highlighting the severe involvement of the DArgic system in late-stage disease and suggesting significant implications for the understanding and treatment of ALS [38]. The use of a D2 DA agonist improved the motoneuronal function in animal models as well as human trials [40,41,42]. The targeting of the DArgic pathway using cerebral DA neurotrophic factors (CDNFs) also seems to be an innovative and effective treatment for ALS, which has shown promise in increasing DA activity via neuroprotective and neurorestorative effects on DA neurons [43].

Since motor neuron degeneration in ALS is not localized [44], the principal aim of DBS when used in ALS patients is to provide symptomatology relief, especially regarding movement-related symptoms, muscle stiffness and tremors.

#### 2.1.2. Huntington’s Disease (HD) Symptomatology and Underlying Mechanisms: Dopaminergic Hyperactivity

Another deadly neurodegenerative disorder, Huntington’s disease (HD), is a hereditary disorder with an early onset, 0–50 years (in [45]) characterized by involuntary choreatic movements in addition to cognitive problems [46]. The progressive loss of neurons has been localized in several brain areas, which could explain the variety of symptoms involving motor, cognitive and even sleep disturbances [47].

Preclinical and clinical studies have shown that the DA system plays a crucial role in HD development. Abnormal DA/glutamate interactions may explain why the cortex and striatum are so vulnerable in HD [48]. Research in HD patients and in rodent genetic models has revealed changes in the DA level and its receptors in the striatum, as well as alterations in DArgic receptor signaling [49] Research in human brains affected by the disease has revealed that in the early stages of progression, DA levels increase, while in advanced stages, they decrease [47]. This increase in early phases may contribute to the involuntary movements characteristic of the disease, known as chorea [50]. Positron emission tomography (PET) studies of HD mouse models show significant reductions in the D2 receptor density in the striatum, cortex and dentate gyrus, even before the onset of overt symptoms [51]. Alterations in DA homeostasis seem crucial in the disease, especially in the motor and cognitive symptomatology, with previously described aberrant DA receptor mediation and DA release even before symptoms are evident [51].

DA stabilizers were proposed as a promising new treatment option for HD [52]. One of the most commonly used pharmacological treatments in HD is the use of anti-dopaminergic agents [53]. Pripodine, a DA stabilizer, via its action on the DA D2 receptor, belongs to a new class of compounds called dopidines, which have been found to regulate motor activity in HD models. In experiments with R6/2 transgenic mice (model of HD), pridopidine has been shown to improve motor function [52]. Depending on the stage of the disease and the increase or decrease in DA levels, pripodine stabilizes the DA levels and brings them to control levels [54]. Preclinical research indicated that pridopidine can normalize motor function by reducing locomotor hyperactivity caused by DA or by increasing low locomotor activity in habituated animals without influencing normal locomotor activity [55]. Tetrabenazine (TBZ) (via the inhibition of monoamine uptake into granular vesicles of presynaptic neurons) is commonly used to reduce brain monoamines and treat chorea in the disease [56,57].

Brain stimulation in HD patients has also been used as a means to restore aberrant signaling associated with the disorder [58,59]. This intervention is expected to impact the aforementioned dysregulated DArgic pathway.

#### 2.1.3. Alzheimer’s Disease (AD) Symptomatology and Underlying Mechanisms: DA and DA Receptor Reduction

AD is the most common form of dementia, being responsible for 60–70% of the diagnosed cases. As a progressive neurodegenerative disorder, it is characterized by an ongoing cognitive decline characterized as mild, moderate or severe. DArgic dysfunction has gained attention in the etiology of AD. Between the well-characterized neuronal loss present in AD, the association of DA and DA receptors levels to AD are still a matter of discussion. There is agreement on altered DA levels, but contradicting evidence points towards both increased and decreased DA levels depending on the patient (reviewed in [60]). However, the authors highlight the more prominent set of data towards hypodopaminergic function (reduced DA and DA receptor levels) in AD. Important sex differences have also been described in regard to DA levels in AD patients, which seems relevant due to the higher risk of AD development in women [61]. A marked decrease in D2 receptors in the hippocampus and frontal cortex were found in AD patients’ brains [62]. The use of pharmacological agents targeting the DA system seems to be beneficial for the patients at different stages of neurodegeneration, with some acting as DA agonists [63,64,65,66], while others seem to act as inhibitors [67,68].

Approved medication to treat AD includes Donepezil, Rivastigmine and Galantamine (cholinergic enhancers) as well as memantine (NMDA antagonist). Interestingly, most of those drugs have shown to increase DA release [69,70]. Lecanemab is the most recently (2023) approved treatment for AD, which has shown to slow the cognitive as well as functional decline in patients at the early stage of disease progression [71]. However, lots of discussion has followed its approval [72].

The relevant role of DA in cognitive processes is undeniable [73,74], as well as how alterations in the DA system correlate with memory disturbances found in AD in clinical and preclinical studies, especially due to its role in the hippocampus [75,76]. Unsurprisingly, the targeting of that brain area and the DArgic system in AD patients has been the focus of interest of many researchers across the years. Stimulation approaches such as DBS and transcranial current stimulation (tACS) show promising beneficial results in AD, although they also have limitations [77,78,79,80].

#### 2.1.4. Parkinson’s Disease (PD) Symptomatology and Underlying Mechanisms: Dopaminergic Loss

PD is the second most common neurodegenerative disorder, currently understood as an interplay of genetic and environmental factors [81]. PD has been related to defects on the direct DArgic pathway, whose main role is to boost movement [82]. PD is characterized by motor disturbances such as tremors and bradykinesia associated with Lewy bodies and the loss of DArgic neurons in the Substantia Nigra pars compacta (SNpc) [83,84].

The primary treatment option for PD focuses on targeting the DA system, aiming to increase its levels in the synaptic cleft. The drug treatment, based on excellence, is L-DOPA, a DA precursor that has been shown to reduce altered motor disturbances in PD [85]. However, it suffers from lots of limitations, such as a quite short effect on the patients and adverse reactions such as dyskinesias [86]. For decades, direct modulation of the DA pathway through the use of agonists on D1, D2 and D5 receptors has proved efficient in alleviating PD symptoms in clinical trials [87,88,89]. Polymorphisms on multiple DA D3 receptors have also been linked to PD susceptibility [90].

Additional pharmacological agents targeting the DArgic system are still booming. Produodopa (acting as a DA precursor) was recently approved in some countries [91] but displays some limitations that are currently slowing down its approval for its use worldwide, which also happens with so many other agents [92]. However, alternative treatments using brain stimulation continue to be explored, focusing on relevant brain areas for motor control (for more detail, see [93,94]).

#### 2.1.5. Drug Addiction: Underlying Mechanism Related to Dopaminergic Implication

Drug addiction processes share altered neurobiological pathways with neurodegenerative disorders. There are genetic, as well as environmental factors, that can contribute to both groups of disorders. DArgic alterations in neurodegenerative disorders and, therefore, treatment focused on boosting DA levels could lead to addictive behaviors. It could increase the likelihood of additional substance abuse or vice versa: drug addiction could enhance vulnerability to neurodegenerative disorders. For example, in PD treatment with DA precursors, patients develop addictions to DA agents due to the rewarding effects perceived after treatment [95]. However, drug abuse, such as under alcohol addiction, has been shown to hasten AD progression [96]. Notably, the impact of environmental circumstances, such as those inducing oxidative stress, have proved to alter the underlying mechanisms common to both conditions, inducing neurodegeneration (for example, [97]) as well as addiction [98]. Therefore, a review on the role of DA in drug addiction is included in this section.

Drug addiction is a chronic psychiatric disorder characterized by compulsive drug-seeking and drug-abusing behaviors. As discussed in previous sections, the DArgic reward system (mesolimbic DA) is a key driver of addiction, and its dysfunction is a defining characteristic of drug addiction. The effects of most drugs of abuse are mediated by large and rapid increases in the level of DA in the Nucleus Accumbens (Nacb). D2 receptors as well as DA signaling have been proposed as biomarkers of drug vulnerability due to their link to impulsivity [99]. Decreases in D2 receptors and DA release have been described in individuals with substance use disorders [100]. A general hypodopaminergic state has also been described after prolonged use of drugs [101]. This reduction in DArgic response is implicated in the neurobiological alterations associated with addiction, contributing to the compulsive drug-seeking behavior and decreased sensitivity to natural rewards observed in these patients. Decreases in DA function have also been associated with reduced regional activity in the orbitofrontal cortex (involved in salience attribution; its disruption results in compulsive behaviors), cingulate gyrus (involved in inhibitory control; its disruption results in impulsivity) and dorsolateral PFC (involved in executive function; its disruption results in impaired regulation of intentional actions) [102]. Additionally, DA transporters (DATs), crucial mechanisms for physiological DA homoeostasis, are the principal target site for multiple psychostimulant drugs, including cocaine and amphetamines [103]. Interestingly, mutations on the DAT gene have been related to vulnerability to drugs such as alcohol [104] or heroin [105].

For most of the addictive drugs, there are currently approved medications to stop or even reduce the intake. There are some promising compounds that have some efficacy reducing craving, intake or relapse (e.g., antatbus for alcohol and methadone or naltrexone for opioids). Even though it is limited, stimulation approaches are being tested for drug addiction [106]. To date, several studies have demonstrated the beneficial effects of DBS or TMS on addictive behaviors and cravings. Notably, studies involving alcohol, opioid and heroin addictions have reported significant reductions in these parameters among patients following DBS treatment [107,108,109,110]. Recent reviews highlight the advances on the use of stimulation devices for drug addiction treatment [111,112]. It has been hypothesized that DBS may alleviate addiction symptoms through normalizing DA levels and restore the functioning of the reward system [113]. Additional studies are necessary to confirm the efficacy of DBS in treating substance use disorders and to establish its long-term safety and effectiveness [114]. 

Figure 1 summarizes the representative alterations related to the DArgic system in some of the neurodegenerative disorders previously described.

### 2.2. Brain Localization of Aberrant Signaling 

After reviewing the primary alterations in neurodegenerative disorders involving DArgic disturbances, it is crucial to identify the specific brain regions where these alterations take place in order to guide treatment strategies. Pharmacotherapy has been instrumental in elucidating the roles of neurotransmitters in these pathologies. However, considering the specific functions of brain regions and their connectivity is also a critical aspect of effective treatment, particularly when employing stimulation approaches.

DArgic cell bodies are located mostly in the midbrain, in the ventral tegmental area (VTA) and Substantia Nigra pars compacta (SNpc), areas that play crucial roles as modulators of their projecting areas. Most of the SNpc neurons project to the cerebral cortex (CC) and the dorsal striatum (referred to as the nigrostriatal pathway). Additional projections from the VTA also connect to the ventral striatum, including the Nacb, and the DArgic pathway can cause a variety of symptoms depending on their origin. Due to the presence of DA across the brain, it is crucial to know each brain area, its functions and the role of DA in order to investigate new targets, thus helping to improve and reverse symptoms.

Clinical trials and preclinical animal models employing brain area stimulation have provided significant insights into the DArgic system, highlighting its importance as a therapeutic target for neurodegenerative and psychiatric disorders.

Despite the innovations mentioned in Section 1, knowledge in regard to the specific role mediated by stimulation approaches in neurodegenerative and psychiatric conditions mostly pertains to the use of DBS. The year 1987 marked a pivotal moment in the understanding of the brain as an electrical circuit, with the first publication reporting the use of the subthalamic nucleus (STN) as a target for alleviating motor symptoms in PD patients. Almost 40 years later, DBS continues to be the main stimulation technique used in clinical settings to treat motor disturbances. Since then, numerous studies have helped provide support for the use of brain stimulation in neurodegenerative and psychiatric disorders. Importantly, recently, an algorithm has been created to generate personalized and symptom-specific DBS treatment plans, which ensures a more beneficial use of DBS [115].

Although initial studies focused on the STN, it is crucial to review the literature to better understand the roles that various brain regions can play as targets for symptom recovery in neurodegenerative and psychiatric disorders. Despite limited supporting data on some of these brain areas, it is important to explore the findings related to them, given their known roles in the symptoms of these pathologies. As discussed, DA is distributed throughout the brain, and disruptions in DA homeostasis have been implicated in neurodegeneration. Therefore, identifying the most effective brain targets could significantly benefit future patients. Deciphering alterations in specific brain signaling using correlational studies could help identify aberrant activities in brain areas, which could serve as reliable biomarkers of individual diseases [116].

#### 2.2.1. Subthalamic Nucleus (STN)

The STN is one relevant area inside the basal ganglia, recognized as a clinical target for treating motor symptoms in disorders such as PD [117]. It continues to be the most common target for stimulation directed to treat neurodegenerative disorders, such as PD [118]. Important for its role in motor control, the STN has projections to the cerebellar cortex [119]. STN DBS also correlated with decreased glucose metabolism in the striatum and thalamus and with increases in metabolism in cortical and limbic cortexes [120], giving support to the importance of the role that these brain areas have in mediating effects on projection sites. The stimulation of the STN influences the DArgic system (Figure 2), for example, by decreasing VMAT2 in additional brain areas such as the caudate, putamen and cortical and limbic regions, areas strongly implicated in movement control [120]. STN DBS, in preclinical studies, resulted in increased survival and firing of DArgic neurons in PD rodent models [121,122,123,124,125], which also led to an increase in the production of DA in other brain areas such as the SNpc, reducing the motor symptoms [126]. An increase and normalization in D1 receptor levels as well as strong decreases in D2 and D3 receptor expression in the Nacb and striatum were also found after STN DBS in PD rodent models [127].

Additionally, STN stimulation modulates DA metabolism by increasing vesicular DA release and increasing the DOPAC/dopamine ratio [128]. Effects of DBS in the STN on DArgic cells in pre-clinical models could involve reduced excitotoxicity and increased dendritic spine density [129]. DBS in the STN has also been considered a promising therapeutic strategy in the treatment of specific addictions where DA levels have been described to be altered; DBS may thus lower the basal level of DArgic neuron activity in these structures and, ultimately, alter the patient’s reactivity to the stimuli [130].

However, some results differ consistently in regard to the impact of STN DBS on the DArgic system, depending on the methods being used.

The level of stimulation is important and should be considered when choosing a brain area to be stimulated. For example, STN DBS significantly reduced the spiking activity of DArgic cells in the SNpc PD pharmacological rat model when using 150 Hz, 60 μs and 400 μA [131] compared to the increased DA firing rate after SNT stimulation in Benazzouz et al.’s work [125], which used 130Hz and 300 μA with a lesion PD model.

There is strong support from clinical trials showing the beneficial effects of STN DBS in patients. A lot has been explored with the stimulation of the STN for PD. STN DBS also improved the gait parameters in PD patients [132]. Specifically, there have been reported improvements in tremors, rigidity, bradykinesia, gait, postural stability and additional activities of daily living in these patients [133]. SPECT images showed that STN DBS in PD patients undergoing simultaneous L-Dopa treatment showed increased postsynaptic striatal D2R availability accompanied by a lack of progression of the motor symptoms [134]. More recently, STN DBS had beneficial effects improving motor symptoms in PD patients, benefits that were further enhanced when combined with DArgic treatment [135]. Importantly, the stimulation of the STN had broader effects on PD patients; aside from motor improvements, cognitive function and mood also increased, with depression scores reduced by 40% [120].

However, targeting the STN is not exclusive to PD studies. STN DBS could decrease the response to cocaine [130]. Moreover, DBS of the STN is also proposed in HD cases where the response to medication does not improve motor symptoms, with efficacy in chorea symptoms [136,137].

#### 2.2.2. Substantia Nigra Pars Compacta (SNpc)

The SNpc is a critical brain region due to its role in the production of DA. The most important function of the SNpc could be its role producing DA and its connections and DA distribution across other brain areas. SNpc releases DA to the striatum, which later on projects to basal ganglia. Additionally, this last brain area connects to the thalamus and motor cortex [138]. Due to the projections connecting SNpc and other relevant structures, it has an important role modulating motor control and coordination mainly. However, the SNpc is also involved in the control of reward processing, which implicates learning as well as motivation aspects.

Alterations of DA in the SNpc have been previously described in neurodegenerative disorders. For example, the loss of DA in the SNpc projections to VTA has been described in mouse models of ALS [38]. It has also been suggested that there is an excess of DA production from the SNpc in HD [139]. Increased numbers of DArgic cells have been found in the SNpc after treatment with tetrabenazine (inhibitor of monoamine uptake) in an HD rat model [57]. One of the main hallmarks of PD is the loss of DA neurons in this area, the main factor of the imbalance of DA levels found on its projection sites in these patients [140]. Pathological lesions in the SNpc were also confirmed decades ago in patients diagnosed with AD, where a loss of neurons was demonstrated [141].

Even though the SNpc has been recognized as a relevant brain area due to its role mediating DA levels, it has mainly been the target of stimulation to recover motor impairments in PD. For instance, DBS of the SNpc showed to have therapeutic effects on PD preclinical models, protecting SN neurons [142] and improving akinesia [143] in PD rat models, as well as in clinical trials where, for example, SN stimulation showed improved gait parameters [127]. Moreover, high-frequency DBS of the SN, more precisely the SN reticulata (SNr), could promote extinction and prevent the reinstatement of methamphetamine-induced CPP in drug addiction [144].

#### 2.2.3. Cortical Cortex (CC)

As the largest brain structure, as well as a highly complex one, targeting the CC could be used to mediate a wide variety of neural functions. The CC contains areas related to sensory and motor control and, additionally, boasts countless connections to other relevant brain areas. Therefore, lots of attention has been set on its possible role as a target of stimulation approaches in neurodegenerative disorders. It is particularly relevant that cortical areas can be directly altered using DBS as well as using stimulation in projected areas such as the STN [145,146].

In the context of ALS, rTMS has been investigated as a potential therapeutic intervention to address changes in cortical excitability and improve some of the symptoms of the disease [147]. A study used magnetic resonance imaging (MRI) to measure grey matter atrophy (cortical and subcortical) in ALS patients. Compared to healthy individuals, ALS patients experience significant grey matter loss that worsens with disease progression, regardless of the rate of progression [110]. This fact gives support to the hypothesis about the beneficial effects of stimulating such areas in these patients. By regulating cortical excitability, motor neuron degeneration can be prevented. Repetitive stimulation may encourage the formation of new neural connections and brain restructuring, which could compensate for the loss of MNs in ALS patients [147]. In the context of addiction, the direct stimulation of the cortex provided important effects on extinction processes as well as on relapse [148,149].

#### 2.2.4. Striatum

Communication from the CC to the subcortical region in the basal ganglia, the striatum, takes place mainly via glutamate. However, this connection is ultimately regulated by DA [150]. The striatal structure plays a crucial role in the coordination of movement [148,151], as well as in various cognitive and emotional functions [152].

Increasing data support targeting the striatum for its beneficial effects in several disorders where DA plays a main role. Compared to the levels observed in healthy individuals, in ALS patients, DA receptor levels were markedly reduced in some brain regions: the Nacb, part of the striatum, the superior frontal gyrus on both sides of the frontal lobe, the left temporal lobe and the angular gyrus region. This reduction was associated with mild cognitive impairment in ALS [153]. Functional and structural changes also occur in the striata of HD patients [57]. Previous studies using striatal DBS show its effect increasing well-being and reducing cravings in patients with addictions [110,154]. Consistently, DA levels are decreased after DBS stimulation of the striatum, more precisely the Nacb [155].

In drug addiction treatment, the Nacb is a primary target site for stimulation, impacting both the reward and the aversion processes, playing a role in neurobiological circuits for withdrawal or cravings [109]. Precisely, Nacb stimulation has shown beneficial effects in the treatment of heroin [156], morphine [157,158], cocaine [159,160], alcohol [161,162,163] and opioid [108] addictions due to its direct impact on DA and DA receptor action.

#### 2.2.5. Pedunculopontine Nucleus (PPN)

The PPN is another relevant brain area of the mesencephalic locomotor system, playing a modulatory effect in many motor and non-motor features [164]. It has been the focus of attention due to its role in axial motor functions, specifically in symptoms such as gait freezing and falls in PD conditions where PPN DBS showed beneficial effects [165]. Its DBS is recommended mostly in severe medication-refractory gait freezing only at low stimulation frequencies.

The stimulation of the PPN has shown to be an effective treatment of postural instability and gait disorders [166]. Dayal et al. [167] summarized relevant cases where PNN stimulation yielded significant results in PD patients’ symptomatologies; however, they describe the relevance of the subtype of the disorder for greater benefits from the stimulation.

PPN stimulation has been considered for other neurodegenerative disorders, such as Multiple System Atrophy (MSA) and Progressive Supranuclear Palsy (PSP) [168,169], due to the prevalence of movement symptoms including gait and postural issues and loss of balance. However, the data supporting its DBS are limited.

#### 2.2.6. Globus Pallidus (GP)

The GP is an additional subcortical structure of the brain, a component of the basal ganglia. This brain region has been targeted for electrical stimulation due to its known role in the control of conscious and proprioceptive movements [170]. So far, DBS of the GP has shown beneficial effects on PD patients, raising from 28 to 64% of the time with good mobility (without the presence of dyskinesia) [133]. Its beneficial effects in PD models have been linked to its effects on striatal DA [171]. Clinical trials on HD patients are also using GP DBS, with encouraging improvements in some motor symptoms present in the course of the disease [clinical trial NCT02535884]. HD patients with late-onset disease also benefit from pallidal DBS, decreasing the choreatic symptoms [58,172,173].

#### 2.2.7. Additional Brain Areas

The dorsal raphe nucleus (DRN), the lateral hypothalamus (LH), the ventral tegmental area (VTA) and the spinal cord (SC) have also been considered for brain stimulation when treating some neurodegenerative disorders as well as drug addiction.

The SC serves as a connection area, whose primary role is to send motor and sensory information between the brain and the rest of the body [174]. The SC has been the target of treatment for chronic pain for decades. However, in preclinical studies, SC DBS improved motor deficits [175,176]. Stimulation of the SC in PD patients with postural instability and gait disorders that have resistance to more standardized treatments have shown positive results and a reduction in their symptoms [177,178]. Moreover, if the stimulation of the SC is maintained, there is an improvement in the Unified Parkinson’s Disease Rating Scale (UPDRS-III) motor scale scores, a reduction of gait episodes and self-reported quality of life.

DArgic fibers projecting to the motor cortex originate from the VTA and are mostly directed to the deep layer of the cortex. The VTA contains mesolimbic and mesocortical DArgic neurons. Therefore, the VTA is really important for its involvement in the reward system, which could be compromised in neurological diseases as well as some psychiatric conditions such as addiction. Abnormalities in the function of VTA DA neurons and the targets they influence are implicated in several prominent neuropsychiatric disorders, including addiction [179]. In stimulation studies involving the VTA, the stimulation typically does not target the VTA directly; instead, researchers study VTA activation resulting from connections with other areas. However, the VTA has been shown to be a relevant target area for stimulation in anxiety and depression studies [180]. Due to the main projections of the VTA, such as the Nacb, the VTA has also been stimulated to check its effects on disorders such as addiction [181]. DBS stimulation of the VTA induced a persistent suppression of Nacb tonic DA levels [182]. Its stimulation was able to suppress reinstatement and seeking behaviors caused by amphetamines [181]. The stimulation of the LH, a brain area involved on the DArgic pathway relevant for reward processing, facilitated extinction of morphine place preference and disrupted drug priming- and stress-induced renewal of morphine place preference [183].

To bring this section to a close, it is essential to recognize the valuable insights gained from these stimulation studies. Nevertheless, significant work remains to accurately target the mechanisms underlying the adverse symptoms associated with neurodegenerative and psychiatric disorders. The promising results offer hope, particularly for patients who do not respond adequately to less invasive pharmacological treatments.

A comprehensive examination of the limitations associated with DBS for the treatment of neurodegenerative disorders is beyond the purview of this manuscript. Nevertheless, Section 4 initiates with a synopsis of these challenges. For a more detailed discourse on the current status and constraints of DBS, readers are directed to the reviews by, for example [129,183,184,185].

## 3. Innovations on Nanomedicine to Target Aberrant Signaling in Neurodegeneration: Focus on the Dopaminergic Pathway

As reviewed in Section 1, current technologies are offering exciting new approaches to treat a broad spectrum of diseases by targeting various brain regions and, consequently, the associated neurotransmitter systems. However, the use of non-invasive ways to treat neurodegenerative disorders remains elusive. Section 2 emphasized the current data on stimulation related to neurodegeneration, which is mostly limited to the use of DBS. Despite advancements, challenges such as device-related complications and medication side effects persist, indicating that significant progress is still required. The pace of knowledge advancement is not as rapid as desired.

When we first heard about Neuralink, people reported this being the start of a new era. Are we in an evolving landscape of innovation or science fiction? Neuralink has placed brain stimulation on the top of the iceberg and has becotme the point of reference for most research laboratories. We have made significant strides in neuroscience; however, knowledge about the brain is far from complete. We presume deep knowledge of pathological mechanisms mediating disorders such as AML, PD and AD. However, we still face several complex and inter-related challenges due to the multifaceted interactions between brain areas as well as between neurotransmitter systems.

Due to our current incomplete understanding of the mechanisms underlying these disorders, the challenge in diagnosis and early detection as well as in drug development keep clinical trials far away from a “magic” treatment. However, there is still great support to research groups that continue to fascinate the audience with promising results and additional innovative approaches.

In the early 2000s, the concept of nanomedicine began to gain significant attention [186]. The research on nanoparticles has, since then, started exploring their use for medical applications, including, but not limited to, targeted drug delivery [187].

### 3.1. Nanotechnology Highlights

Thanks to nanoformulation and conjugation, we have gained enhanced drug solubility and bioavailability due to a nanoparticle’s ability to encapsulate hydrophobic drugs. This achievement has therefore ensured a higher amount of a pharmacological agent to reach the brain. Most nanoparticles are designed to overcome Blood–Brain Barrier (BBB) penetration limits. The BBB serves as a highly selective interface, limiting the permeability of substances from the bloodstream into the central nervous system (CNS) and thereby presenting a formidable challenge for drug delivery to the brain. Nanomedicine offers a promising approach to overcome this barrier through the use of nanoparticles, which can be engineered to improve drug solubility, protect therapeutic agents from degradation and facilitate targeted delivery to specific brain tissues. This innovative strategy enhances the efficacy and precision of treatments for neurological disorders. Additionally, as previously mentioned, the functionalization of nanoparticles with targeting ligands (which can include specific peptides or even antibodies with a high affinity) could reduce off-target effects, minimizing unwanted side effects. Importantly, nanoparticles can also be developed to provide sustained therapeutic effects, therefore reducing the need of frequent, repeated dosing. Several laboratories have focused on the use of temperature-responsive nanoparticles (for example, hydrogel nanoparticles) that allow for the release of conjugated compounds in response to temperature changes, therefore, allowing on-demand drug release in the brain [188].

### 3.2. Nanotechnology for DArgic Target in Neurodegeneration and Addiction

The main aspects of innovative nanomaterials are their surface properties and surface functionality, which play relevant roles in the operational characteristics of such materials. A new era is now being exploited focusing on such nanoparticle characteristics.

There have been few efforts to target the DArgic system with the use of nanoparticles. Due to DA relevance in several disorders (as previously discussed), Kook et al. [189] designed magnetic nanoparticles to specifically target DA molecules and, thus, used them to quantify DA physiological levels, highlighting its possible role as an indicator for early disease diagnosis. As mentioned, nanotechnology is evolving and allows for surface modification of the pertinent nanoparticles for specific conjugation and, therefore, in this case, DA targets. Recent efforts to target the DA system with magnetic nanoparticles also showed the potency of nanoparticle surface modification using DA derivatives in in situ polymerization to produce bioactive nanocomposites [190]. These authors highlight the reduced spontaneous release of the DA derivatives from the nanoparticles due to the high stability of the synthesized coatings. Mayeen et al. [191] produced nanocomposites functionalized with DA, which enhanced the electrical and magnetoelectric properties of their BaTiO_3_ nanoparticles.

On the other hand, a growing body of evidence explored the use of nanoparticles to directly target the DArgic system via nanoconjugation with specific pharmacological agents, with encouraging results. For example, nanoparticles directly conjugated with DA [192] or conjugated with L-Dopa to enhance its delivery for the treatment of PD showed promising results in improving symptoms in PD models. These studies highlight the increased delivery and release of the DA precursor using polymeric nanodelivery systems [193]. These investigations reveal better therapeutic efficacy and extended L-Dopa release [194,195]. Several types of nanoparticles have also been used to increase the DA levels in AD models [196,197]. Other manipulations using nanoparticles, for example, mesoporous silica nanoparticles, have been used to convert fibroblasts into DArgic neuron-like cells, which could be a relevant approach for neurodegenerative diseases including ALS [198]. 

However, current knowledge does not limit the use of nanoparticles for the aforementioned disorders. For example, while the application of nanoparticles specifically for addiction treatment is still emerging, their potential to deliver precise doses of DA-regulating agents could be significant for future therapies aimed at modulating DA levels to treat addiction. Because some addictive drugs such as cocaine and amphetamines exert their effects via a direct impact on DA transporters (DATs), new nanomaterials aimed to target DATs are being considered.

Does nanomedicine go in parallel to the brain stimulation approach? Could we benefit from both treatment options? As reviewed in Section 2, integrating nanomedicine with targeted testing in specific brain regions known to mediate motor, cognitive and other critical processes altered in neurodegenerative disorders presents a promising approach.

## 4. Introduction of Upcoming Technologies for Brain Stimulation: Magnetoelectric Nanoparticles, the Missing Link between Nanoformulation and Stimulation Approaches

### 4.1. Current Limitations

Even though such innovative nanoparticle approaches discussed in Section 3 are encouraging and are providing new ways to target neurodegenerative disorders, they present relevant limitations that require attention. Briefly, it is important to note that crossing the BBB continues to be a handicap on drug efficacy, even with the use of some innovative nanoparticles. Additionally, due to the complexity and heterogeneity of the brain, achieving precise delivery into the brain areas of interest (due to their individual relevance in the aforementioned disorders previously discussed) is challenging with current nanoparticles. Also, controlling the adequate rate of release represents a major issue in this field. As with other biochemical compounds, it is always a concern for long-term biocompatibility due to the possibility of potential toxicity in the long term. Even though we recognize the tremendous potential of current nanoparticles in development, we are still far from the “divine” treatment. All these aspects could limit the clinical use of currently available nanomaterials [199,200].

Revising the previously discussed stimulation approaches (Section 1 and Section 2 above), we face the same situation: we have promising results but are surrounded by numerous limitations. We would like to highlight that we encounter a tremendous challenge: the full knowledge of the underlying mechanisms by which the “electric” brain works, and therefore, the mechanism by which all stimulation approaches can affect those neural circuits. Now, we briefly summarize the relevant barriers of current stimulation devices. In most of the stimulation cases introduced across this review, the invasiveness that they represent limits their use to severe cases where the anticipated benefits could outweigh the risks of such intervention. The need for surgical implantations involves several possible complications (infections, hardware-related complications, bleeding, etc.), which additionally brings the difficulty for accurate targeting and, therefore, precise stimulation of the brain area of interest. Due to the side effects present in the patients, in the short term as well as long term, they are still not totally described. Additionally, some devices that use magnetic induction or optoelectronic signaling have strong limitations related to tissue penetration [201,202,203]. This incomplete knowledge calls for additional work focused on how the brain works and how we can improve its functioning via neural stimulation. Due to the variability observed in response to stimulation interventions, nowadays, it is still really complicated to confirm that a patient would benefit from such treatment. And, finally, it is extremely relevant to mention that access to these technologies remains restricted in so many countries due to the high cost derived from the expenses of the surgical procedure itself, among other costs.

### 4.2. The Progress and Evolution of the New Era: Magnetoelectric Nanoparticles (MENPs)

Given our current understanding and comprehensive grasp of the strengths and constraints inherent in existing treatment modalities, we are strategically positioned to progress and innovate, leveraging novel tools to effectively surmount prevailing challenges.

In pursuit of cutting-edge technologies to surpass current scientific limitations, over the past decade, Dr. Khizroev pioneered the development of magnetoelectric nanoparticles (MENPs), often referred to as “magic tools”, originally outlined in a computational study [204]. Following that initial study, a decade of research supports the innovative use of MENPs for their application in medical science. MENPs are recognized as a non-invasive alternative to present treatment approaches with a potential for widespread therapeutic use in a variety of psychiatric and neurodegenerative diseases [205]. There are currently numerous laboratories engaged in this advanced nanotechnology worldwide. From our knowledge, a few research groups located around the world (Spain, Germany, Switzerland, Russia, Korea, China, and the USA) are now optimizing the composition and use of MENPs for targeting the brain. Are MENPs the “magic” bullet we have been looking for? Achieving targeted drug release upon request coupled with precise wireless onsite stimulation for on-demand control represents the envisioned solution.

Briefly, MENPs could overcome most of the limitations of current nanotechnology as well as stimulation approaches. MENPs can generate electric fields through the application of magnetic fields, allowing for their use as non-invasive devices for biomedical applications that can include precise brain stimulation upon request as well as targeted drug delivery as desired [206]. The detailed characteristics of MENPs and their potential in medicine have recently been reviewed [205,206,207,208]. Among the relevant aspects to overcome, MENPs would not produce heat, as generated by other devices, thus eliminating high temperatures as a source of off-target neuromodulation [209,210]. More specifically, MENPs can address the limitations of current nanomedicine related to nonspecific drug release prior to reaching target sites. MENPs can be precisely directed to specific locations, thereby preventing premature drug release or undesired effects on non-target brain regions. Additionally, compared to traditional stimulation devices, MENPs offer the advantage of being non-invasively guided to targeted sites without the need for surgical intervention or the associated complications of current stimulation techniques. MENPs can be wirelessly stimulated, ensuring targeted activation while avoiding the stimulation of unintended brain areas. Moreover, MENPs enable the integration of both drug release and stimulation approaches. This dual functionality allows for a synergistic therapeutic strategy, combining targeted pharmacological treatment with localized stimulation, thereby enhancing the overall efficacy and precision of the intervention.

During the last few years, we are gathering strong evidence supporting the ability of MENPs to induce a neuronal response. Under physiological conditions, a neuron’s maximum firing rate is typically 500 Hz [211]. In response to wireless magnetic stimulation, MENPs increase the neuronal calcium channel response in vitro and in vivo [209,210,211,212,213,214] comparable to conventional wired DBS. The use of several frequencies below the approved conventional stimulation parameters used in clinical trials with DBS (<500 Hz), at frequencies that vary between 10 and 20 Hz [212], 100 Hz [214] and 140 Hz [209,210], have been shown to stimulate neurons non-invasively and without inducing neuronal damage or inflammation.

### 4.3. Magnetoelectric Nanoparticles (MENPs) for DArgic Target

Ongoing research has shown the ability to locate MENPs in the brain. Crossing the BBB with MENPs was initially achieved by exposing rodents to a magnetic field after intravenous MENP administration [215]. Later, magnetic resonance imaging (MRI) was used to achieve the effective delivery of intravenously administered MENPs to the brain in baboons [216]. It was the first time that behavioral parameters were considered when studying MENP effects in vivo. More recently, we showed the direct capability of MENPs to reach the brain via intranasal administration [217]. With no doubt, at doses tested, MENPs seem to be safe and non-toxic when located in the brain [209,210,211,212,213,214,215,216,217,218,219].

However, research on how MENPs can directly affect several underlying pathways that control a patient’s behavior is still at initial stages of study. We previously discussed the relevant role of specific brain areas on behavioral control and, precisely, the DArgic system study on that modulation (see Section 2 above). It was not until 2021 that Kozielski’s group [209,210] targeted a specific brain area (STN) with MENPs. Importantly, these studies showed that bilateral STN injection of MENPs does not induce detrimental effects on rodents’ behavior. As previously discussed, the STN is an important brain area regarding DArgic implication in neurodegenerative disorders, such as PD and HD, that involve motor deterioration. Therefore, it was not surprising that magnetic stimulation of the STN at frequencies shown to induce neuronal modulation (140–280 Hz) induced locomotor activity in MENP (average size 250 nm)-treated animals [209,210]. Additional work from our group (in preparation) also showed the ability of MENPs, when directly administered in motor areas, to induce behavioral activation in rats. We show the ability of MENPs to induce motor response in a msec range after stimulation. Additionally, we demonstrate micron spatial resolution of wirelessly induced MENP local stimulation. Therefore, even though current studies are limited, the impact of MENP stimulation on motor control is irrefutable, and its role on the DArgic system seems unquestionable. Alosaimi et al. [210] confirmed that the stimulation of the STN using MENP induced additional changes in neuronal activity in projection areas such as the motor cortex (MC) and the paraventricular region of the thalamus (PV thalamus). They confirmed that STN stimulation modulated the neuronal activity of the mesolimbic VTA DArgic neurons. More precisely, TH cells’ activity was changed (but not cell count) after MENP stimulation [210]. Previously, MENP magnetic stimulation of the STN also led to higher neuronal activity in projection areas from the cortico-basal ganglia-thalamocortical circuit, including the motor cortex and thalamus [209].

There are significant points to discuss regarding these recent studies: the MENPs used by the Kozielsi group did not migrate from the site of administration for up to 7 weeks post-injection. It is challenging to determine whether this is advantageous or disadvantageous. On one hand, it could be considered beneficial, as it allows for the repeated stimulation of MENPs without the need of more invasive administration. In this respect, MENPs would closely mimic the function of DBS (device related), which involves an “initial” administration and positioning and long-term sustained use. Conversely, previous studies using intravenous and intranasal administration showed that MENPs dispersed and were eliminated from the system within days to weeks. Additionally, it is important to note that the nanocomposition of MENPs can vary, leading to slight differences between laboratories.

Figure 3 summarizes the potential of MENPs for the treatment of neurodegenerative disorders based on current knowledge of these innovative nanoparticles.

## 5. Conclusions: Peculiarity of the DArgic System and Its Target Using MENPs

DA is unequivocally a critical target for the treatment of neurodegenerative diseases. How can we target a neurotransmitter that is everywhere and has an essential role in brain functioning, avoiding undesired effects? The nature of DA itself represents a challenge. With its five main pathways, each of them with different roles in human behavior, to target on-demand DA only present in a specific pathway, in a specific brain area, is a difficult obstacle to overcome. DArgic neurons respond to electrical stimulation (see Section 2). We previously revised how the stimulation of different specific brain areas could differ in their relevance to restore brain circuitry in neurodegenerative disorders, with some emphasis on DA modulation.

DA depletion or aberrant functioning plays a relevant role in neurodegenerative as well as several psychiatric disorders. DA dysregulation and imbalance in neurodegenerative and psychiatric disorders can be mediated by DAT dysregulation. DAT is a vital mediator controlling DA levels, and its dysfunction has closely been linked to disease. Preserving DArgic neurons and their proper regulation is crucial to prevent disease progression involving motor and non-motor (cognitive) symptomatologies.

Therefore, MENPs’ ability to be guided to specific brain areas where DA has been described to play an important role allows for targeted therapy in a way not previously possible. MENPs’ ability to stimulate neurons and their direct role in neurotransmission has just started to be tested [210]. Because MENPs have shown their ability to be conjugated with specific antibodies or pharmacological compounds [220,221,222,223], the hypothesis of MENPs for specific DA targeting is not questionable. The precision previously described about these nanoparticles allows for designing MENPs to specifically deliver agents with neuroprotective and neurorestorative characteristics directly to neurons where, for example, DA receptors are located. MENPs could be conjugated with agents like L-DOPA or DAT inhibitors to prolong DA action to, for example, compensate for DA loss in PD. This approach at early stages of neurodegeneration could, per se, reverse the progression of aggressive disorders as previously introduced. Current funded projects are evaluating MENPs’ role in rodent models of PD; therefore, it is just a matter of time to give additional support to the hypothesis mentioned.

Nanometer-size neural stimulators can be seamlessly and fully implanted into specific brain areas through stereotaxic injections, therefore leveraging the benefits of nanomaterials with minimal invasiveness [214]. To conclude, MENPs represent an innovative approach far away from current treatment interventions. MENPs can be used as nanocarriers as well as stimulation devices. As seen with DBS, where medication in addition to DBS showed additive effects in improving motor performance in PD [224], MENPs’ beneficial effects are promising. Previous and ongoing studies confirm the reliable biocompatibility, biosafety and feasibility of MENPs. Therefore, as recently discussed, a new era is ensured, where the side effects of current pharmacological (including nanomedicine) and stimulation approaches could be minimized with less invasive proposals. With the increasing amount of data regarding brain heterogeneity and its functions, the close future is encouraging. We emphasize individualized treatments, with in vivo real-time MRI for MENP localization and, therefore, non-invasive on-demand stimulation, with, if desired, additional drug release into the selected brain area.

## 6. Materials and Methods

The current review has been prepared using the Pubmed, Embase and Scopus databases. We searched for the following keywords: neurodegeneration, dopamine, Alzheimer, Parkinson, Huntington, sclerosis, addiction, nanoparticles, magnetoelectric nanoparticles, neuromodulation, subthalamic nucleus, globus pallidus, striatum, pedunculopontine nucleus, cortex, substantia nigra, deep brain stimulation, transcranial magnetic stimulation, and brain-computer-interfaces.

## Figures and Tables

**Figure 1 molecules-29-03580-f001:**
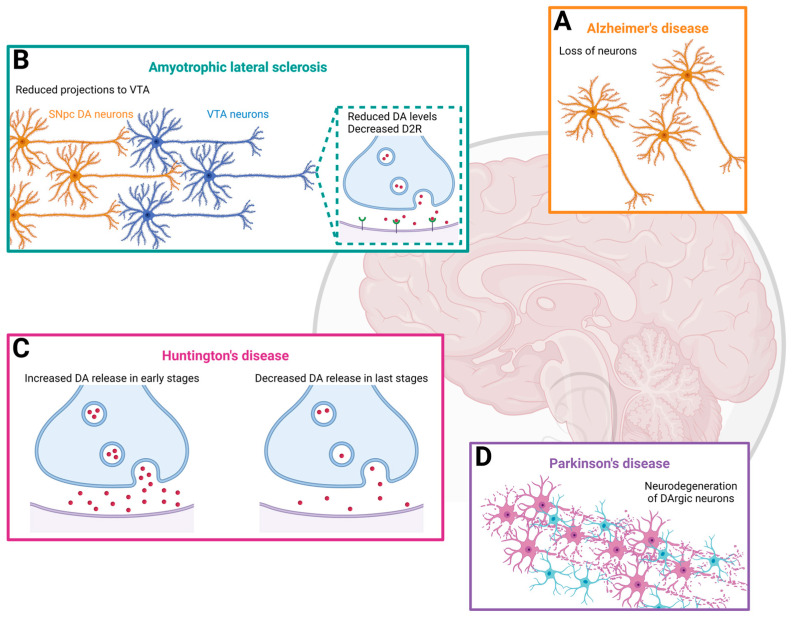
Dopaminergic alterations in several neurodegenerative disorders. (**A**) Neuronal loss detected in Alzheimer’s disease; (**B**) reduced projections from DArgic neurons of SNpc to VTA, with additional reduced DA levels and decreased D2 receptors in Amyotrophic Lateral Sclerosis; (**C**) altered DA homeostasis, with increased and reduced DA levels in early and late stages, respectively, of Huntington’s Disease; (**D**) neurodegeneration of DArgic neurons in the Substantia Nigra pars compacta in Parkinson’s Disease. Created with BioRender (https://www.biorender.com/ (accessed on 17 July 2024)).

**Figure 2 molecules-29-03580-f002:**
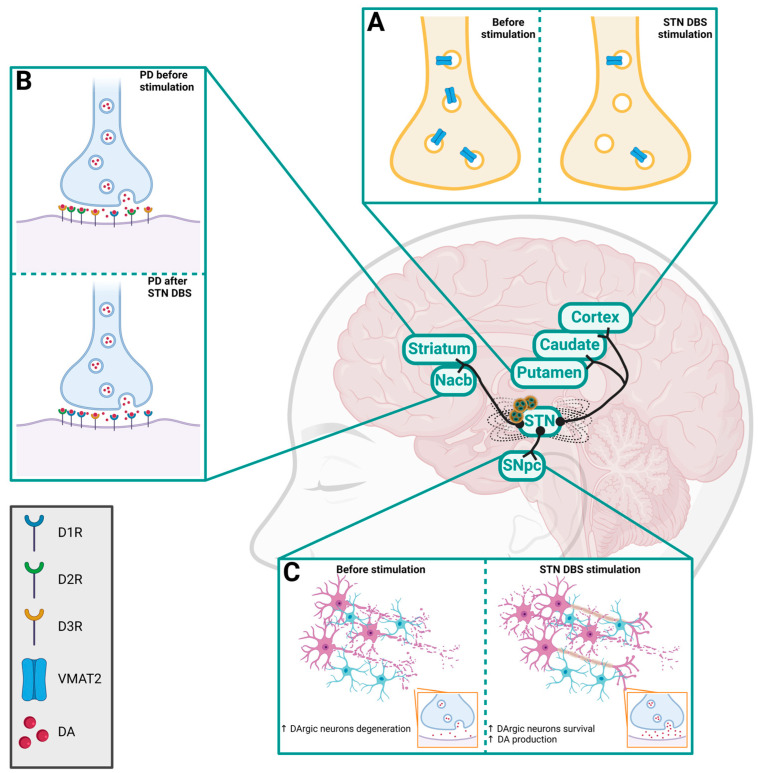
Effects of STN DBS on the DArgic projection sites. (**A**) Reduction of VMAT2 in caudate, putamen and cortex; (**B**) increase in D1 receptor levels and decrease in D2 and D3 receptor numbers in the Nacb and striatum; (**C**) increases in survival of DArgic neurons and DA production. Created by BioRender (https://www.biorender.com/ (accessed on 17 July 2024)).

**Figure 3 molecules-29-03580-f003:**
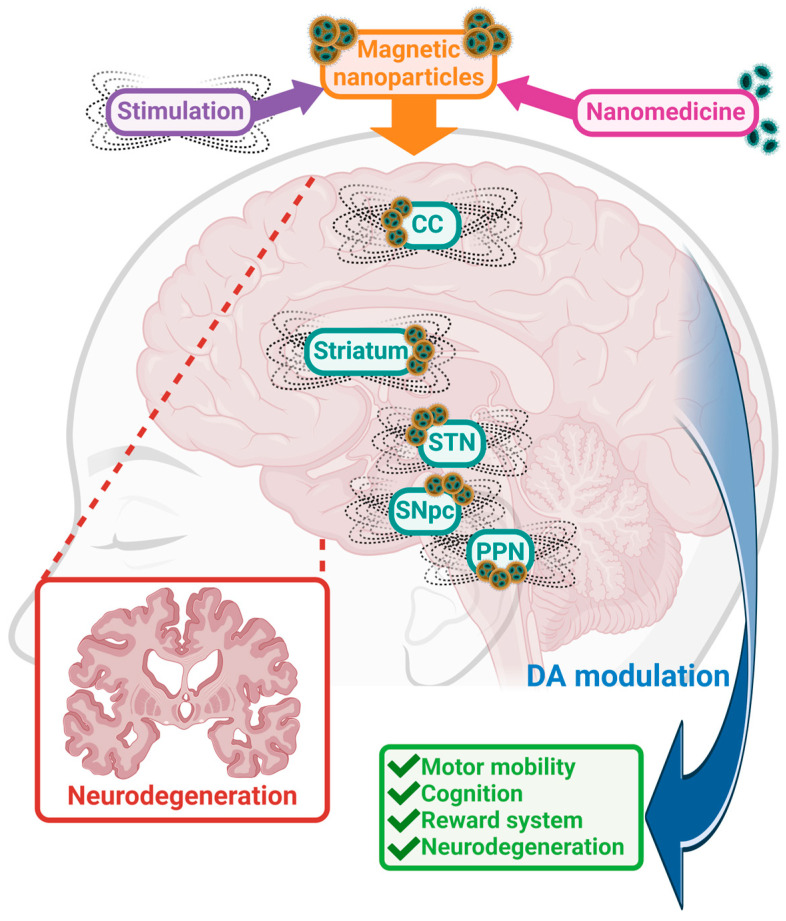
Schematic representation of the effects of MENPs on the neurodegenerative brain. MENPs can be externally directed towards specific brain areas (Cortical Cortex (CC); striatum; subthalamic nucleus (STN), Substantia Nigra pars compacta (SNpc), Pedunculopontine Nucleus (PPN)). Upon localization, MENPs can provide targeted stimulation to modulate the DArgic system, thereby alleviating the pathological symptoms associated with neurodegenerative diseases. Created with BioRender (https://www.biorender.com/ (accessed on 18 April 2024)).

## Data Availability

No new data were created or analyzed in this study.

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
