# Peer review of "Advances in Brain Stimulation, Nanomedicine and the Use of Magnetoelectric Nanoparticles: Dopaminergic Alterations and Their Role in Neurodegeneration and Drug Addiction"

_molecules, 2024, doi:10.3390/molecules29153580_

Round 1

Reviewer 1 Report

Comments and Suggestions for Authors

The manuscript has discussed the advancements of brain stimulation via either invasion stimulator or non-intrusive pharmacological region-targeting medicine. Authors also talked about these techniques been utilized in different diseases. I only have minor comments:

The abstract emphasizes the importance of the non-invasive technique. While the section 1 spend lot of words on intrusive stimulation, and section 2 start to talk about the non-intrusive way of treatment. This way of presentation may cause confusion of the readers to have difficulty to catch the main point of the manuscript. I suggest either (1) add words of deep brain stimulation in the abstract, or (2) switch section 1 and section 2 to firstly discuss pharmacological-based techniques, then talk about invasive technique.

In the legend of Fig. 1, there is a typing error “localyzation” (page 15 of 26, line 746), which should be “localization”.

In page 17 of 26, line805, the words “Please add: ” can be removed.

Author Response

Reviewer 1

Comments and Suggestions for Authors

The manuscript has discussed the advancements of brain stimulation via either invasion stimulator or non-intrusive pharmacological region-targeting medicine. Authors also talked about these techniques been utilized in different diseases. I only have minor comments:

The abstract emphasizes the importance of the non-invasive technique. While the section 1 spend lot of words on intrusive stimulation, and section 2 start to talk about the non-intrusive way of treatment. This way of presentation may cause confusion of the readers to have difficulty to catch the main point of the manuscript. I suggest either (1) add words of deep brain stimulation in the abstract, or (2) switch section 1 and section 2 to firstly discuss pharmacological-based techniques, then talk about invasive technique.

  • Answer to reviewer: We appreciate reviewer´s comment. Following the suggestion, we have modified the abstract as follows. Line 19 now states: “Deep Brain Stimulation (DBS) is an FDA-approved treatment for specific neurodegenerative disorders, including Parkinson's Disease (PD), and is currently under evaluation for other conditions, such as Alzheimer's Disease. This technique has facilitated significant advancements in understanding brain electrical circuitry by enabling targeted brain stimulation and providing insights into neural network function and dysfunction. In reviewing DBS studies, this review places particular emphasis on…”.

In the legend of Fig. 1, there is a typing error “localyzation” (page 15 of 26, line 800), which should be “localization”.

  • Answer to reviewer: It has been corrected.

In page 17 of 26, line 859, the words “Please add: ” can be removed.

Answer to reviewer: It has been corrected.

Reviewer 2 Report

Comments and Suggestions for Authors

1- The title: I suggest the authors rethink about modifying the title to include also "magnetoelectric nanoparticles.  

2- The review is too lengthy with overcrowded data that may make the reader confused. The article needs re-organization and rearrangement of data into: experimental and clinical

3- The section of DBs: please include the challenges concomitant to DBS. The section of addiction needs to present some clinical studies in this field and their results as a line of treatment for addiction.

4- In the section of nanotechnology and nanomedicine, please add some knowledge about the BBB obstacle in passage of drugs and the concept and advantages  of nanomedicine as a drug delivery system in this situation.

5- In the section of magnetoelectric nanoparticles: please rewrite the section clarifying how this technique overcome limitations in DBS or nanoparticle approaches.

Comments on the Quality of English Language

The quality of English language is accepted

Author Response

Reviewer 2

Comments and Suggestions for Authors

1- The title: I suggest the authors rethink about modifying the title to include also "magnetoelectric nanoparticles.  

  • Answer to reviewer: We thank reviewer suggestion. The title has been slightly modified as follows: “Advances in Brain Stimulation, Nanomedicine and the use of Magnetoelectric Nanoparticles: Dopaminergic alterations and its role in Neurodegeneration and drug addiction”.

2- The review is too lengthy with overcrowded data that may make the reader confused. The article needs re-organization and rearrangement of data into: experimental and clinical

  • Answer to reviewer: We appreciate reviewer´s comment. We acknowledge the complexity of the text. Due to current status of the use of stimulation devices, the majority of the data focusing on DA and specific brain areas is experimental. To answer reviewer concern, we have added the following on the text: Line 153 “Several brain stimulation devices are approved for therapeutic applications in neurodegenerative conditions. DBS devices, including those from Medtronic, Abbott (formerly St. Jude Medical), and Boston Scientific, are used to treat PD, essential tremor, and dystonia. Vagus Nerve Stimulation (VNS) systems such as the VNS Therapy System by LivaNova are approved for epilepsy, which can sometimes be associated with neurodegenerative disorders. While TMS devices like the NeuroStar TMS Therapy system and Brainsway Deep TMS system are approved for depression, obsessive compulsive disorder (OCD), and migraine, they are being explored for potential applications in neurodegenerative diseases like AD and PD. Responsive Neurostimulation (RNS) systems such as the NeuroPace RNS System are approved for epilepsy, with ongoing research into their efficacy for other neurodegenerative conditions. These devices undergo stringent regulatory scrutiny by bodies such as the U.S. FDA to ensure their safety and effectiveness in managing neurodegenerative disorders. Therefore, most of the data to be collected and discussed in the following sections pertains to experimental conditions. Limited clinical data, where available, is also presented for comprehensive analysis”

3- The section of DBs: please include the challenges concomitant to DBS.

Answer to reviewer: We have revised the DBS section (section 2). Following reviewer comment, we have added the following sentence on our manuscript: line 571 “A comprehensive examination of the limitations associated with DBS for the treatment of neurodegenerative disorders is beyond the purview of this manuscript. Nevertheless, Section 4 initiates with a synopsis of these challenges. For a more detailed discourse on the current status and constraints of DBS, readers are directed to the reviews by, for example, Davidson et al. (2024a, 2024b) and Yaseri et al. (2024).”

Those 3 references have been added on the reference list.

188.Davidson B, Milosevic L, Kondrataviciute L, Kalia LV, Kalia SK. Neuroscience fundamentals relevant to neuromodulation: Neurobiology of deep brain stimulation in Parkinson's disease. Neurotherapeutics. 2024a Apr;21(3):e00348. doi: 10.1016/j.neurot.2024.e00348. Epub 2024 Apr 4. PMID: 38579455; PMCID: PMC11000190.

189.Davidson B, Vetkas A, Germann J, Tang-Wai D, Lozano AM. Deep brain stimulation for Alzheimer's disease - current status and next steps. Expert Rev Med Devices. 2024 Apr;21(4):285-292. doi: 10.1080/17434440.2024.2337298. Epub 2024 Apr 4. PMID: 38573133.

  1. Yaseri A, Roozbeh M, Kazemi R, Lotfinia S. Brain stimulation for patients with multiple sclerosis: an umbrella review of therapeutic efficacy. Neurol Sci. 2024 Jun;45(6):2549-2559. doi: 10.1007/s10072-024-07365-3. Epub 2024 Jan 30. PMID: 38289559.

Limitations or challenges of DBS were currently introduced in section 4.1, Current limitations, as follows: line 679” Revising the previously discussed stimulation approaches (section 1-2 above), we face the same situation, we have promising results but are surrounded by numerous limitations”. …. “Let's briefly summarize relevant barriers of current stimulation devices. In most of the stimulation cases introduced across this review the invasiveness that they represent limits their use to severe cases where the anticipated benefits could outweigh the risks of such intervention. The need for surgical implantations involve several possible complications (infections, hardware-related complications, bleeding, etc), which additionally brings the difficulty for accurate targeting and, therefore, precise stimulation of the brain area of interest. Due to the side effects present in the patients, short-term as well as long-term, they are still not totally described. Additionally, some devices that use magnetic induction or optoelectronic signaling have strong limitations related to the tissue penetration [197-199]. This incomplete knowledge calls for additional work focused on how the brain works and how we can improve its functioning via neural stimulation. Due to the variability observed in response to stimulation interventions, nowadays it is still really complicated to confirm that a patient would benefit from such treatment. And, finally, it is extremely relevant to mention that the access to these technologies remain restricted in so many countries due to the high cost derived from expenses of the surgical procedure itself between other costs”.

Given the focus of this review on current advancements and innovations, a detailed discussion on the limitations of DBS is not included, as it falls outside the scope of this manuscript. Numerous reviews have thoroughly evaluated DBS concerning neurodegenerative disorders. To illustrate, several recent reviews  from 2024 have addressed the benefits and limitations of DBS:

  1. Davidson et al. (2024a) investigate the controversial aspects of DBS in Parkinson's disease (PD).
  2. Davidson et al. (2024b) provide a comprehensive summary of the safety, feasibility, and outcomes of DBS, discussing topics for further research, such as optimization of electrode placement for its use in Alzheimer's disease (AD).
  3. Yaseri et al. (2024) assess the effectiveness of brain stimulation in multiple sclerosis, identifying limitations and areas requiring further research.

4-The section of addiction needs to present some clinical studies in this field and their results as a line of treatment for addiction.

  • Answer to reviewer: Following reviewer´s request, we have added the following information on the text: line 342 “To date, several studies have demonstrated the beneficial effects of DBS or TMS on addictive behaviors and cravings. Notably, research involving alcohol, opioid and heroin addiction has reported significant reductions in these parameters among patients following DBS treatment (Bach et al., 2023; Kuhn et al., 2014; Taremian et al., 2019; Chen et al., 2019). Recent reviews highlight the advances on the use of stimulation devices for drug addiction treatment (Iqbal et al., 2023; Mehta et al., 2024).”.
  1. Bach P, Luderer M, Müller UJ, Jakobs M, Baldermann JC, Voges J, et al. Deep brain stimulation of the nucleus accumbens in treatment-resistant alcohol use disorder: a double-blind randomized controlled multi-center trial. Transl Psychiatry. 2023;
  2. Kuhn J, Möller M, Treppmann JF, Bartsch C, Lenartz D, Gruendler TO, et al.: Deep brain stimulation of the nucleus accumbens and its usefulness in severe opioid addiction. Mol Psychiatry 19:145–146, 2014
  3. Taremian F, Nazari S, Moradveisi L, Moloodi R. Transcranial direct current stimulation on opium craving, depression, and anxiety: a preliminary study. J Ect. 2019;35:201.
  4. Chen L, Li N, Ge S, Lozano AM, Lee DJ, Yang C, et al. Long-term results after deep brain stimulation of nucleus accumbens and the anterior limb of the internal capsule for preventing heroin relapse: an open-label pilot study. Brain Stimul. 2019;12:175–83.
  5. Iqbal J, Mansour MNM, Saboor HA, Suyambu J, Lak MA, Zeeshan MH, Hafeez MH, Arain M, Mehmood M, Mehmood D, Ashraf M. Role of deep brain stimulation (DBS) in addiction disorders. Surg Neurol Int. 2023 Dec 22;14:434. doi: 10.25259/SNI_662_2023. PMID: 38213452; PMCID: PMC10783698.
  6. Mehta DD, Praecht A, Ward HB, Sanches M, Sorkhou M, Tang VM, Steele VR, Hanlon CA, George TP. A systematic review and meta-analysis of neuromodulation therapies for substance use disorders. Neuropsychopharmacology. 2024 Mar;49(4):649-680.

5- In the section of nanotechnology and nanomedicine, please add some knowledge about the BBB obstacle in passage of drugs and the concept and advantages  of nanomedicine as a drug delivery system in this situation.

  • Answer to reviewer: Thank you for the suggestion. We have added the following information on the text in section 3.1: line 607 “The BBB serves as a highly selective interface, limiting the permeability of substances from the bloodstream into the central nervous system (CNS) and thereby presenting a formidable challenge for drug delivery to the brain. Nanomedicine offers a promising approach to overcome this barrier through the use of nanoparticles, which can be engineered to improve drug solubility, protect therapeutic agents from degradation, and facilitate targeted delivery to specific brain tissues. This innovative strategy enhances the efficacy and precision of treatments for neurological disorders”

6- In the section of magnetoelectric nanoparticles: please rewrite the section clarifying how this technique overcome limitations in DBS or nanoparticle approaches.

  • Answer to reviewer: Thank you for the comment. We have modified the section with the following clarifications to highlight how MENPs can overcome current limitations: line 727 “More specifically, MENPs can address the limitations of current nanomedicine related to nonspecific drug release prior to reaching target sites. MENPs can be precisely directed to specific locations, thereby preventing premature drug release or undesired effects on non-target brain regions. Additionally, compared to traditional stimulation devices, MENPs offer the advantage of being non-invasively guided to targeted sites without the need for surgical intervention or the associated complications of current stimulation techniques. MENPs can be wirelessly stimulated, ensuring targeted activation while avoiding the stimulation of unintended brain areas. Moreover, MENPs enable the integration of both drug release and stimulation approaches. This dual functionality allows for a synergistic therapeutic strategy, combining targeted pharmacological treatment with localized stimulation, thereby enhancing the overall efficacy and precision of the intervention”.
